# Management of Drug Resistance in Mantle Cell Lymphoma

**DOI:** 10.3390/cancers12061565

**Published:** 2020-06-12

**Authors:** Gaël Roué, Brigitte Sola

**Affiliations:** 1Lymphoma Translational Group, Josep Carreras Leukaemia Research Institute (IJC), 08916 Badalona, Spain; 2MICAH Team, INSERM U1245, UNICAEN, CEDEX 5, 14032 Caen, France

**Keywords:** B-cell lymphoma, cyclin D1, proteasome inhibitor, ibrutinib, NF-kB pathway, mutation, innate resistance, acquired resistance, combinatory treatment, therapeutic strategy

## Abstract

Mantle cell lymphoma (MCL) is a rare but aggressive B-cell hemopathy characterized by the translocation t(11;14)(q13;q32) that leads to the overexpression of the cell cycle regulatory protein cyclin D1. This translocation is the initial event of the lymphomagenesis, but tumor cells can acquire additional alterations allowing the progression of the disease with a more aggressive phenotype and a tight dependency on microenvironment signaling. To date, the chemotherapeutic-based standard care is largely inefficient and despite the recent advent of different targeted therapies including proteasome inhibitors, immunomodulatory drugs, tyrosine kinase inhibitors, relapses are frequent and are generally related to a dismal prognosis. As a result, MCL remains an incurable disease. In this review, we will present the molecular mechanisms of drug resistance learned from both preclinical and clinical experiences in MCL, detailing the main tumor intrinsic processes and signaling pathways associated to therapeutic drug escape. We will also discuss the possibility to counteract the acquisition of drug refractoriness through the design of more efficient strategies, with an emphasis on the most recent combination approaches.

## 1. Physiopathology of Mantle Cell Lymphoma

Mantle cell lymphoma (MCL) is a rare B-cell lymphoma that represents 5–10% of all non-Hodgkin lymphomas (NHLs), with an incidence of 0.8 cases per 100,000 persons [1]. It develops primarily among elderly individuals with a median age of approximately 67 years and a male-to-female ratio of 2–3:1. At diagnosis, 70% of patients or more have disseminated disease (stage III or IV), with lymphadenopathy (75%), hepato-splenomegaly (35–60%), bone marrow (>60%) and peripheral blood (13–77%) involvement [2]. Waldeyer’s ring and extranodal sites including the gastrointestinal tract, are also frequently involved [3]. The clinical evolution is usually very aggressive, and despite overall response rates above 70% with standard immunochemotherapeutic schemes (see Section 1.3), few patients can be cured [4].

### 1.1. MCL Subtypes

MCL has been recognized as an aggressive small B-cell lymphoma that developed in a linear fashion from naive B-cells. Paradoxically, a subset of patients follows an indolent clinical evolution with a stable disease, and a longer survival, even in the absence of chemotherapy [5], reflecting, in part, that MCL develops along two different pathways. Classical MCL (cMCL) is usually composed of *IGHV*-unmutated or minimally mutated B-cells that express SOX11 (SRY (sex determining region Y)-box 11), features genetic instability and typically involves lymph nodes and other extranodal sites. Acquisition of additional molecular/cytogenetic abnormalities can lead to even more aggressive, blastoid or pleomorphic MCL. Leukemic non-nodal MCL (nnMCL) develops from *IGHV*-mutated SOX11−B cells, carrying epigenetic imprints of germinal center (GC)-experienced B cells. It usually involves peripheral blood, bone marrow, and often spleen. These cases feature genetic stability and are frequently clinically indolent; however, secondary abnormalities, often involving *TP53*, may occur and lead to a very aggressive disease. A third MCL subtype, in situ mantle cell neoplasia (ISMCN), is characterized by the presence of cyclin D1+ cells; most typically in the inner zone of the follicles. Although disseminated, this subtype appears to have a low rate of progression (Figure 1) [3]. Morphologically, three main subtypes of MCL are recognized: the classic, the blastic/blastoid, and the pleomorphic variants. The last two subtypes have higher proliferation rates and are associated with inferior clinical outcome [5].

### 1.2. MCL Biological Features and Prognostic Factors

The phenotype of MCL is relatively characteristic with high expression of IgM/IgD surface immunoglobulins. Immunophenotyping reveals that neoplastic cells are usually CD5+ and CD43+ and express the B-cell-associated antigens CD19, CD20, CD22, and CD79. They are usually negative for CD3, CD23, CD11c, CD10, and CD200. MCL cells are generally B-cell lymphoma (BCL)-2 positive and BCL-6 negative [7]. Demonstration of t(11;14)(q13;q32) by FISH or cyclin D1 overexpression by immunohistochemistry is generally required to diagnose MCL, although a small number of cases are cyclin D1−. These cases have a high expression of cyclin D2 or cyclin D3; however, this is not helpful for diagnostics as these proteins are also overexpressed in other B-cell neoplasms. The nuclear SOX11 expression is a highly specific marker for both cyclin D1+/− MCL [8]. SOX11 is a transcription factor that has been reported to block terminal B-cell differentiation by regulating *PAX5* expression in aggressive MCL. There is also data demonstrating a role for SOX11 as a driver of pro-angiogenic signals in MCL through the regulation of platelet-derived growth factor A, contributing to a more aggressive phenotype [9].

A specific MCL international prognostic index (MIPI) classifies MCL patients into low, intermediate, and high-risk groups, based on four independent prognostic factors: age, Eastern Cooperative Oncology Group (ECOG) performance status, lactate dehydrogenase (LDH), and leukocyte count [10,11]. Other factors such as proliferation of the tumor, karyotypic complexity, genetic aberrations, and DNA methylation are independent prognostic factors for MCL outcome [12].

### 1.3. MCL Therapy

Some newly diagnosed MCL patients can be diligently observed, deferring therapy to a later date. Asymptomatic, low tumor burden MCL cases with non-nodal presentation and genetic stability are candidates for this strategy [13]. Delayed treatment in these patients does not adversely affect overall survival (OS) from time of treatment initiation [14]. Although the monoclonal antibody (mAb) anti-CD20 rituximab is considered a standard of care for all newly diagnosed MCL patients, for patients requiring frontline therapy, the initial therapeutic decision is dictated by the age and the fitness of the patient. Since the 1990s, a standard regimen of cyclophosphamide, hydroxydaunomycin (doxorubicin), vincristine, and prednisone (CHOP) has been frequently used to treat MCL patients. Response rates associated with CHOP in this disease are rarely complete or durable, compared with those observed in other B-cell aggressive lymphomas. Therefore, more-intensive strategies have been explored, combining additional agents to improve both the response rates and the durations of response. Induction regimens have included rituximab and high-dose cytarabine (araC) (an antimetabolite pyrimidine analogue), usually followed by autologous stem cell transplantation (ASCT) in younger patients (see below) [15]. The addition of rituximab to CHOP (R-CHOP) was further established as a standard-of-care regimen for the treatment of naive MCL patients. This regimen is now typically administered to patients who are elderly and considered intermediate to high risk, as well as those with relapsed or refractory (R/R) disease, and has been associated with improved OS [16]. However, median survival remains around 5 years, and it is not yet entirely clear how the improved outcomes observed in clinical trial have translated to real-world settings. For patients that achieve remission, consolidation therapy is recommended [17]. For older, less-fit patients there is no generally accepted frontline therapy. R-CHOP regimen followed by rituximab maintenance achieved a significant improvement of OS, with a 4-year survival rate of 87%, largely superior to the 63% survival obtained with interferon (IFN)α therapy [18].

In transplant-ineligible patients with untreated, newly diagnosed MCL, a phase 3 trial demonstrated that frontline bortezomib plus rituximab, cyclophosphamide, doxorubicin, and prednisone (VR-CAP regimen) was associated with a survival benefit over R-CHOP, with a median OS of 90.7 months, significantly longer that the value observed in the R-CHOP group (55.7 months). Therefore, this approach should be considered as a standard of care in this subgroup of patients [19].

Maintenance therapy with rituximab after R-CHOP-based induction has demonstrated clear survival benefit in MCL patients, therefore it represents a well-established approach for postponing disease progression. Among novel agents, the thalidomide-derivative, immunomodulatory drug (IMiD), lenalidomide (Revlimid), has not demonstrated benefit when used as maintenance therapies in MCL, while the first-in-class Bruton’s tyrosine kinase (BTK) inhibitor, ibrutinib (Imbruvica^®^) is still under investigation in these settings (see Section 2.4) [17].

While ASCT is preferentially used in youngest/fit cases as first-line consolidation treatment and almost never employed in the real-cohort patients in R/R MCL [20], allogeneic stem cell transplantation (alloSCT) produces long-term disease-free remissions for around 30–40% patients, especially in younger patients with early relapse or MCL refractory to induction therapy. This approach is considered the sole potentially curative therapy for R/R MCL [21]. In front-line settings, alloSCT was demonstrated to be feasible but should only be considered for patients at high risk of early progression following conventional therapy [22].

Due to the limitations of stem cell transplantation and also considering the relatively poor outcomes associated with chemotherapy, the potential for several chemotherapy-free strategies has been evaluated in MCL patients since early 2000s. Consequently, a growing number of biologically-targeted therapies are profoundly altering the landscape of MCL treatment options in both first-line and relapsed settings [17]. Among these agents, there are currently four drugs licensed across the world: the proteasome inhibitor bortezomib (BTZ, Velcade^®^), the mechanistic target of rapamycin (mTOR) inhibitor temsirolimus (Torisel^®^), lenalidomide, and ibrutinib. As single agents, overall response rates (ORRs) are 33% (8% complete response (CR)), 22% (2% CR), 28% (8% CR), and 68% (21% CR), respectively [23,24,25,26]. Beside this clinical efficacy, major differences have been observed in the degree and frequency of adverse events (AEs) associated to these agents in MCL patients. In bortezomib-receiving patients, the most commonly reported AEs are asthenia (72%), peripheral neuropathies (55%), constipation (50%), diarrhea (47%), nausea (44%), and anorexia (39%), the apparition of neuropathy being the most common toxicity, leading to discontinuation and eventually to death [27]. In the case of temsirolimus, hematological toxicity (thrombocytopenia, 72–100%; anemia, 52–66%; neutropenia, 24–77%) is the most frequent AE observed in the clinical setting, and can be generally successfully managed by dose reductions or treatment delay [28]. Hematologic toxicity was also the most common AE observed in R/R MCL patients receiving lenalidomide, with neutropenia and thrombocytopenia observed in 40–62% or 28–12% of the cases, depending on the cohort. Importantly, these effects did not culminate into serious events in any studies, all hematological toxicities being manageable and reversible upon discontinuation of the IMiD [29]. Finally, ibrutinib is by far the safest agent among this list, with hematologic AEs limited to thrombocytopenia (22%), neutropenia (19%), and anemia (18%). Other common AEs including diarrhea (54%), fatigue (50%), nausea (33%), dyspnea (32%), and infection (<10%) were mostly observed during the first 6 months of therapy and with less frequency, thus confirming the safety profile of ibrutinib in R/R MCL [30].

Several novel agents using different target points have also been used with some reported efficacy in R/R MCL. Among the most exciting recent advances in the management of B-cell malignancies, has been the development of chimeric antigen receptor (CAR) T cells [31]. In a recent phase 2 study involving MCL patients who did not respond to BTK inhibitor therapy, the anti-CD19 CAR T-cell therapy, KTE-X19, achieved durable remission in patients with R/R MCL (93% ORR, 67% CR, with progression-free survival (PFS) of 61% and OS of 83% at 1 year) but not without risks: many study participants experienced high-grade cytopenias, infections, and neurologic events [32]. Cyclin-dependent kinase (CDK)4/6 inhibitors (e.g., abemaciclib, palbociclib) are also an attractive therapeutic option given the role of cell-cycle deregulation in the pathogenesis of MCL [33]. Anti-CD20 mAbs, such as ofatumumab [34] and obinutuzumab, [35] have single-agent activity in rituximab-treated patients and are good candidates to be used in combination with other therapies (see Section 4.2). Moreover, BH3 mimetic-type BCL2 inhibitors such as ABT-199 (venetoclax), phosphatidylinositol 3-kinase (PI3K)δ inhibitors such as idelalisib, histone deacetylase (HDAC) inhibitors (e.g., vorinostat, abexinostat or panobinostat), mTOR inhibitors (e.g., everolimus), or other small molecules including some second-generation BTK inhibitors, are being developed and explored in MCL [36,37,38,39,40]. Finally, the promising activity of anti-CD38 mAb, such as daratumumab in multiple myeloma, has prompted the initiation of studies in other B-cell malignancies, including MCL [41].

## 2. Molecular Signatures of MCL

### 2.1. Translocation t(11;14)(q13;q32)

Such as for other aggressive lymphomas, the genomic landscape of MCL shows large variations among patients [42,43]. However, in the vast majority of the cases, tumor cells are characterized by the translocation t(11;14)(q13;q32) which juxtaposes the *CCND1* gene encoding cyclin D1 with an enhancer of the Ig heavy chain (*IGH*) gene [5]. This translocation leads to the expression of cyclin D1 that is physiologically never expressed in the B-cell lineage. Associated with its specific kinases, CDK4/6, cyclin D1 phosphorylates and inhibits the retinoblastoma protein (RB1), allowing the release of transcription factors of the E2F family, the transcriptional activation of genes controlling DNA synthesis, and the G1-to-S phase progression within the cell cycle [44]. Consistent with this cell cycle regulatory function, tumor cells overexpressing cyclin D1 display an uncontrolled proliferation. An elevated cyclin D1 expression and in turn, a proliferation signature, are associated with chemoresistance and a reduced MCL patients survival [45]. Cyclin D1 expression is required for MCL cells survival since siRNAs targeting *CCND1* lead to enhanced apoptosis [46] and sensitivity towards drugs [47].

The translocation t(11;14) is considered as the primary oncogenic event but secondary genomic alterations including somatic mutations are necessary for the progression of the disease and contribute to its heterogeneity [5].

### 2.2. Recurrent Genomic Mutations

With the use of recent technologies such as next generation sequencing (NGS) technology, whole-genome sequencing (WGS), whole-exome sequencing (WES), and RNA expression profiling, recurrent genomic mutations associated with MCL have been described. Although, the percentage of mutations varies among the cohorts, the most represented abnormalities concern ataxia-telangectasia mutated (*ATM*), *CCND1*, and *TP53* (encoding the tumor suppressor p53) genes (Table 1). Mutations of *ATM* affect the functional domains of the tumor suppressor protein that signals DNA damage, whereas mutations of *CCND1* are found predominantly in the exon 1, leading to the stabilization of the protein [46]. These mutations are differentially distributed among MCL subtypes according to the *IGVH* and *SOX11* status. Although both are described as oncogenic drivers, they have no prognostic values [48]. The role of ATM in MCL is still debated [49]. In sharp contrast, *TP53* mutations identify a distinct and aggressive form of MCL patients with poor or low response to upfront treatments [50,51] and a shorter OS [48]. *NOTCH1* mutations are also associated with a poor OS [52]. The other characterized mutations target anti-apoptotic genes (*BIRC3*, *TLR3*), cell cycle regulatory genes (*RB1*), genes coding for chromatin-modifier enzymes (*MLL2*, *MLL3*, *WHSC1*, and *MEF2B*) (Table 1).

The comparison of mutations across lymphomas shows that *CCND1*, *RB1* are found mutated exclusively in MCL, whereas *WHSC1*, *ATM*, and *BIRC3*, although prominently altered in MCL, are also found mutated in other B-cell lymphomas [42]. As a consequence of this mutational landscape, MCL cells are highly dependent on cell cycle (*CCND1*, *RB1*), DNA damage response (*ATM*, *TP53*), and NF-κB pathways for their survival, allowing the identification of several putative therapeutic targets among these signaling axes.

### 2.3. Deletions of INK4A/ARF (CDKN2A) Locus

Several studies have investigated the global chromosomal alterations in MCL by comparative genomic hybridization (CGH) array and single nucleotide polymorphism (SNP) array. Frequent alterations in MCL include gain of chromosomes 3q, 7q, and 8q as well as loss of chromosomes 1p, 6q, 8p, 9p, 9q, 11q, and 17p [59,60,61]. The region 9p21 contains the *CDKN2A* locus that codes for two tumor suppressors, p16^INK4A^ and p14^ARF^. p16^INKA^ is the physiological inhibitor of cyclin D1/CDK4/6 complexes whereas p14^ARF^ interacts and sequesters murine double minute 2 (MDM2, the E3 ubiquitin ligase of p53), controlling its degradation through the ubiquitin-proteasome system (UPS). Deletion of this locus is observed in 20% of MCL patients and associated with a proliferation signature [45]. Interestingly, *CDKN2A* can be silenced by BMI1, a polycomb family protein which is overexpressed in some cases [62]. Beside *CDKN2A* loss, the methylation of *CDKN2A* and *CDKN2B* promoters, that leads to the transcriptional repression of these genes, has been described in a subset of MCL patients [60]. Quantitative multiplex PCR analyses confirmed the loss of *CDKN2A* (31% of cases) and *RB1*, *ATM* and *TP53* in 38%, 24%, and 10% of the cases, respectively; *TP53* loss correlated with an unfavorable outcome [63].

### 2.4. Abnormalities of Signaling Pathways

Somatic mutations described in MCL cells lead to the constitutive activation of various signaling pathways. While deregulated p16^INK4A^/CDK4/RB1 and p14^ARF^/MDM2/p53 axes are the most common hallmarks of MCL, several signaling pathways may be overactivated by the chronic activation of cytokine/interleukin (IL) receptors, and/or cell/cell or cell/extracellular matrix interactions promoted by an MCL-specific tumor microenvironment (TME).

#### 2.4.1. B-Cell Receptor Signaling

The B-cell receptor (BCR) consists of Ig chains bound to CD79a/b co-receptors (Figure 2). Upon binding of cognate antigen by the hypervariable regions of the BCR, the LYN tyrosine kinase phosphorylates the intracellular domain of CD79a/b chains (ITAMs) that are then able to recruit the spleen tyrosine kinase, SYK. Once recruited, SYK undergoes a phosphorylation at Tyr-130 residue and consequently activates kinases and adaptor proteins to form the signalosome. The signalosome, which include BTK, activates several downstream pathways governing crucial intracellular events like gene transcription, mRNA translation, cell proliferation, and survival (Figure 2) [64]. Mechanistically, once the signalosome is formed, activated BTK phosphorylates phospholipase (PL)Cγ2 which in turn phosphorylates protein kinase C (PKC)β and caspase recruitment domain-containing protein 11 (CARD11), a key regulator of NF-κB signaling (Figure 3). The transmembrane protein CD19 is also phosphorylated by LYN after BCR triggering and recruits phosphatidylinositol 3-kinase (PI3K) to the BCR. PI3K regulates the phosphorylation of PIP2 (phosphatidylinositol 4,5-biphosphate) to generate PIP3 (phosphatidylinositol 3,4,5-triphosphate). This latest has the capacity to recruit downstream proteins like AKT and BTK at the inner plasma membrane. In turn, AKT is efficiently activated and BTK activity is amplified. Following these events, activated PLCγ2 hydrolyzes PIP2 in diacylglycerol (DAG) and inositol triphosphate (IP3), this latest being involved in the control of intracellular calcium flux, which indirectly activates the nuclear factor of activated T-cells (NFAT). PLCγ2 also promotes the activating phosphorylation of ERK (extracellular signal-regulated kinase) 1/2 (Figure 2) and, to some extent, contributes to the regulation of JNK (Jun NH2-terminal kinase) and p38 MAPK (mitogen-activated protein kinase) pathways (not schematized). As a whole, BCR signaling culminates in the activation of NF-κB, MAPK, PI3K, NFAT pathways, which all promote the proliferation and the survival of B cells.

MCL is characterized by a highly distinctive Ig gene repertory and a biased BCR, suggesting a crucial role for antigenic selection in the pathogenesis of at least a subset of MCL [66]. Indeed, increased BTK autophosphorylation at Y223 has been observed in unstimulated primary MCL cells, together with a high expression of the kinase [67]. A pro-survival role of BCR signaling is suggested by the constitutive phosphorylation of different kinases of this pathway, including LYN, SYK, and PKCβ, observed in a limited panel of patients [68,69]. MCL cells also harbor a constitutive activation of NF-κB and AKT, which might reflect both BCR or toll-like receptor (TLR) constitutive activation [5,70]. Early studies in relapsed setting showed that the BTK inhibitor ibrutinib achieved response rate and CR of 77% and 33%, respectively [71].

#### 2.4.2. NF-κB Signaling

BCR signaling is found activated simultaneously with canonical or non-canonical NF-κB signaling pathways (these are mutually exclusive) in MCL malignant B cells as well as in different components of the TME (Figure 3) [72]. After formation of the signalosome, the phosphorylation of CARD11 by PKCβ allows the formation of the CBM complex containing the BCL10 and MALT (mucosa-associated lymphoid tissue) adaptors (Figure 3). Once the CBM is formed, the IκB kinase (IKK) complex is activated and can phosphorylate NF-κB-inhibitor alpha (IκBα), allowing to the ubiquitylation of this latest and to its degradation by the UPS. Consequently, p50 and RELA transcription factors are released and translocated to the nucleus where they can control gene transcription (Figure 3). The non-canonical NF-κB pathway is triggered by the activation of tumor necrosis factor α (TNFα) receptor (TNFR), B-cell activating factor receptor (BAFFR), and CD40 signaling (Figure 3) [73]. Non-canonical NF-κB signaling leads to the release of p52 and RELB transcription factors (Figure 3) [72]. This pathway depends on the degradation of the p100 precursor and NIK (NF-κB-inducing kinase). When this alternative pathway is inactive, NIK is constantly degraded through the E3 ubiquitin ligase activity of TRAF3 in complex with TRAF2 and cellular apoptosis inhibitors (cIAP1/2). As mentioned previously, recurrent mutations of *TRAF2* and *BIRC3*, two negative regulators of NIK, have been found in 15% of MCL [54]. Thus, constitutive activation of the non-canonical NF-κB pathway in the corresponding patients may identify an MCL subgroup potentially responsive to NIK inhibitors.

#### 2.4.3. TLR Signaling

Different TLRs have been found overexpressed in MCL cells, suggesting that their signaling may be particularly relevant for MCL pathogenesis and tumor progression [74,75]. Binding of TLRs by their cognate ligands triggers the activation of NF-κB (Figure 3), ERK1/2, and AKT pathways, leads to an upregulation of cyclin D1, and consequent enhanced proliferation. Suggesting a functional cross-talk between BCR and TLR signaling, a high level of TLR was associated with a hyper-responsiveness of the BCR machinery and an enhanced expression of genes associated with the NF-κB pathway in MCL cells [76].

#### 2.4.4. PI3K/AKT/mTOR Signaling Pathway

Chronic activation of the PI3K/AKT signaling axis has been found in approximatively one third of patients with classical MCL, while 100% of the blastic/blastoid cases analyzed showed a constitutive activating phosphorylation of AKT at Ser-473 residue. One possible mechanism of activation identified was a loss of the PTEN (Phosphatase and TENsin homolog deleted on chromosome 10) tumor suppressor [77,78]. Beside this genetic loss of PTEN, both chronic, BTK-mediated, and ligand-independent, tonic, BCR signaling can activate PI3K (Figure 2) [64]. Finally, ROR1, a tyrosine kinase-like orphan receptor highly expressed in MCL, binds to CD19 and this complex can activate PI3K/AKT as well as MEK/ERK1/2 pathways [79].

## 3. Molecular Mechanisms of Resistance to Standard/Current Therapeutics

Such as for most lymphomas, the majority of MCL patients respond to initial therapies but often relapse due to the development of drug resistance [80]. Although de novo or primary resistance is mostly carried by gene abnormalities and tumor cells/TME interactions, acquired or secondary resistance to therapeutic drugs requires the reprogramming of the cells and the reactivation of key signaling pathways.

### 3.1. Resistance to BTK and PI3K Inhibitors

As commented above, BCR signaling is constitutively activated in MCL, mediated by activating phosphorylation of BTK at Tyr-223. However, in some cases, the constitutive phosphorylation affects BTK downstream effectors like LYN, SYK, and PKCβ kinases (Figure 2) [81]. Supporting the notion that BTK is indispensable to B-cell and lymphoma survival [82], the targeting of BTK with the irreversible inhibitor ibrutinib has shown promising responses in R/R MCL. Mechanistically, ibrutinib induces lymphocytosis and lymph nodes shrinkage, due to the decrease of interactions between tumor cells and their TME. Nonetheless, almost one-third of MCL patients are resistant to ibrutinib therapy, and sensitive patients eventually acquire resistance, experiencing a more aggressive disease [26,83,84]. The second FDA-approved BTK inhibitor, acalabrutinib, potentially more “kinase-selective”, showed durable response in patients with R/R MCL as single agent [85,86]. However, a longer follow-up is needed to conclude on acalabrutinib efficacy.

Intrinsic resistance to ibrutinib is due, in part, to the activation of the alternative non-canonical NIK-NF-κB pathway (Figure 3) [54]. In line with this observation, *TRAF2* and *BIRC3* are found mutated in ibrutinib-resistant cells [54], leading to NIK accumulation and conferring dependency on this kinase, that could offer a novel target for therapy for ibrutinib-resistant patients. Mutations of *CARD11* have been also observed in MCL patients at relapse after ibrutinib treatment, albeit at a low frequency (5%) [55].

Beside the genetic alterations underlying ibrutinib refractoriness, TME has been shown to mediate de novo ibrutinib resistance through the secretion of BAFF and the activation of both canonical and non-canonical NF-kB pathways [87]. By combining kinomic analysis in vitro and in vivo, Zhao and coworkers recently identified a PI3K/AKT/mTOR-ILK (integrin β1-linked kinase) axis as a central hub for TME-MCL tumor cell interactions for both innate and acquired resistance. MCL cells can develop de novo resistance through a dynamic interplay between lymphoma cells and their TME, mediated by the triggering of BTK, ERK, and AKT activation and enhanced survival through the synthesis of several chemokines and cytokines, including BAFF. Then, cells can acquire another degree of ibrutinib-resistance through the reprogramming of their kinome, the enhanced expression of integrin β1, and the activation of an integrin β1/ILK (integrin-linked kinase) pathway. Interestingly, integrin β1 can form a complex with ILK and mTORC2, generating a positive loop of activation [88].

Despite these different hypotheses, resistance mechanisms may be more complex since ascribed to mutations and/or adaptive mechanisms such as activation of alternative pathway or reprogramming of cell cycle [57,89]. As commented previously, ibrutinib targets irreversibly the BTK, through its ability to bind a cysteine residue (C481). A missense (C481S) point mutation at the ibrutinib binding site confers resistance by preventing drug binding [57,90]. In parallel, *BTK*, *ATM*, and *TP53* mutations were also recorded for MCL patients who discontinued ibrutinib and developed blastoid transformation [91].

Among PI3K proteins, the isoform p110δ is expressed uniquely on hematopoietic cells. Given the constitutive activation of the PI3K/AKT/mTOR in MCL, the targeting of p110δ has been explored mediated by a specific inhibitor, idelalisib. However, the loss of PTEN or the amplification of the PI3K catalytic subunit p110α [77], impair full idelalisib activity [92,93,94]. Importantly, the blockade of p110α enhances the expression of activation-induced cytidine deaminase (AID) and increases somatic mutation and chromosomal translocation frequency and, in turn, genomic instability. The same observation was made although, to a lesser extent, with ibrutinib [95]. Given that ibrutinib is currently being used for the treatment of R/R MCL patients, these data may question its administration for long periods.

Large-scale genomic studies have identified a hotspot for recurring somatic mutations in exon 1 of *CCND1* [42,43]. The most frequent mutations (E36K, Y44D, and C47S) lead to modifications of the C-terminal part of cyclin D1 and accumulation of the protein through a defective proteolysis by the UPS. Moreover, those *CCND1* mutations contribute to ibrutinib resistance although this mechanism is still unknown [46]. Importantly, among the downstream targets of PI3K/AKT are the catalytic partners of cyclin D1, CDK4 and CDK6, the inhibition of which can modify MCL cell cycle and reprogram the cells toward a re-sensitization to p110δ inhibition [57].

Recently, it has been shown in preclinical models that a paradoxical metabolic reprogramming toward oxidative phosphorylation (OXPHOS) can confer ibrutinib resistance [96]. The comparison of ibrutinib-sensitive and -resistant tumor cells by RNA-Seq indicated that differentially expressed genes were related to glycolytic metabolism, tricarboxylic acid cycle, and glutamine transport. The upregulation of MYC and mTORC1 reprograms the metabolism toward OXPHOS by activating genes involved in glycolysis, glutaminolysis, and mitochondrial biogenesis. The upstream effectors of MYC and mTORC1 activation are not known but could be related to cell cycle dysfunctions [96].

Considering the interplay between the PI3K/AKT and BCR signaling pathways (Figure 2) together with the hyperactivation of the PI3K/AKT pathway in cases resistant to BTK inhibitors, combination therapies based on the capacity of p110α inhibition to overcome TME-induced ibrutinib resistance [97], have been tested with some success in preclinical settings [98].

### 3.2. Resistance to Bortezomib and Proteasome Inhibitors

Over the past years, proteasome inhibition has been demonstrated to be an effective therapeutic strategy in MCL. BTZ was the first proteasome inhibitor (PI) approved by the FDA in 2006 as a second-line treatment for MCL patients. However, more than half of patients are either de novo resistant or develop secondary BTZ resistance along the course of the treatment [99]. The development of new generation PIs such as carfilzomib (CFZ) and ixazomib has not completely solved the problem of resistance. We have previously reviewed in details the mechanisms of BTZ resistance in MCL and multiple myeloma [100]. We will focus here on the more prominent aspects of both de novo and secondary PI resistance.

In MCL cells, BTZ innate resistance has been linked to the accumulation of the anti-apoptotic protein MCL-1 [101] and/or to the constitutive activation of the NF-κB signaling pathway [102]. In this last study, NF-κB activation was consequent to a proteasome-independent degradation of IκBα (Figure 3) [102]. The accumulation of the serine/threonine kinase casein kinase 2 (CK2), acting both on NF-κB and signal transducer and activator of transcription 3 (STAT3) survival pathways contributes also to BTZ resistance [103]. The redox status has also been reported as a crucial mediator of BTZ efficacy. Indeed, BTZ induces the generation of reactive oxygen species and the upregulation of the pro-apoptotic NOXA protein [104]. The upregulation of NOXA is impaired in BTZ-resistant MCL cells, with a major role of nuclear factor NF-E2 p45-related factor 2 (NRF2) in this phenomenon [105]. BTZ-sensitive MCL cells display an increase in the expression of NRF2 target genes upon treatment with the drug, whereas resistant cells show minimal variations in this gene signature. Accordingly, an elevated expression of NRF2 target genes at the basal level, predicts a poor sensitivity to PIs [105]. Among the aberrantly activated pathways in MCL, canonical Wnt signaling has been associated with the expression of the zinc finger E-box binding homeobox 1 (ZEB1) transcription factor, responsible for the activation of proliferation-associated genes such as *CCND1*, *MYC*, and *MKI67* and anti-apoptotic genes, including *MCL1* and *BCL2* [106,107]. In turn, ZEB1 level may be considered as a predictive biomarker of BTZ response.

Secondary BTZ resistance results from successive steps along the evolution of the disease and is obviously multifactorial. The activation of the UPR (unfolded protein response) and ER (endoplasmic reticulum) stress pathways in MCL cells exposed to BTZ is required to elicit NOXA transcription [108], and defective UPR regulation consequent to the overexpression of the ER chaperone protein, BiP, is associated with both innate and acquired refractoriness to BTZ [109]. In MCL cells, BTZ leads to the intracellular accumulation of both anti-apoptotic MCL-1 and BH3-only protein, NOXA. By interacting with MCL-1, NOXA allows the release of the pro-apoptotic effector, BAK, leading to mitochondrial depolarization and initiation of the apoptotic cascade [101]. Interestingly, the inhibition of cyclin D1/CDK4 activity in MCL cells reduces the stabilization of NOXA, directing the protein through degradation by an autophagy mechanism [110].

Resistance to BTZ has also been associated with plasmacytic differentiation of MCL cells. Using BTZ-adapted cell lines, Pérez-Galán and coworkers showed that these cells express some plasmacytic features including interferon regulatory factor 4 (IRF4) upregulation as well as CD38 and CD138 expression, but not other B-cell differentiation hallmarks such as Ig secretion or X-box-binding protein 1 (XBP1) splicing [99]. Further studies associated this phenotype with an increased tumorigenicity of MCL cells in in vivo settings [111]. Interestingly, SOX11, which is overexpressed in a majority of MCL cells, is the master regulator for the shift of a mature B-cell into a plasmacytic phenotype [112]. The silencing of SOX11 downregulates PAX5, induces BLIMP1, upregulates IRF4 and promotes B-cell differentiation. Of note, BLIMP1 is also a mediator of NOXA-induced apoptosis in MCL and is required for BTZ-induced apoptosis in MCL cell lines and primary samples [113].

### 3.3. Resistance to Lenalidomide

As previously commented, lenalidomide has shown some efficacy in R/R MCL patients including those resistant to BTZ [25,114,115]. The antitumor activity of lenalidomide and other IMiDs is mediated through their direct effects on the immune cells (T and NK cells) present in the TME, on the TME itself by modulating inflammatory cytokines, and by indirect effects on malignant B cells. In particular, preclinical studies have shown that lenalidomide enhanced NK cell-mediated cytotoxicity against MCL cells, by promoting the formation of lytic immunological synapses and the secretion of granzyme B [116,117]. Nonetheless, lenalidomide also interacts with the E3 ubiquitin ligase cereblon (CRBN) expressed in MCL patients, and enhances its activity to degrade zinc-finger transcription factors IKZF1 (Ikaros) and IKZF2 (Aiolos), and to decrease IFR4 activity [118]. As detailed before, the hyperactivation of the IRF4/MYC axis is associated with BTZ resistance. In turn, lenalidomide cooperates with the BET bromodomain inhibitor CPI203, an indirect inhibitor of MYC transcriptional program, to overcome BTZ resistance [111]. Associated with dexamethasone, lenalidomide can also target STAT3 and PI3K/AKT pathways, both indirectly involved in BTZ resistance [119]. Upon lenalidomide treatment, cyclin D1, as a downstream target of these two pathways, is downregulated and dissociated from the CDK inhibitor, p27*KIP1*, and may thus account for this sensitizing effect of lenalidomide [120]. Despite these advances, the intrinsic mechanisms of lenalidomide resistance in MCL remain only partially known. Among described mechanisms are the upregulation of MCL-1, the downregulation of BAX, and the activation of PI3K/AKT signaling pathway consequently to the interference of the hypoxic TME with NK cell-mediated cytotoxicity [121]. These potential mechanisms are supported by genetic alterations affecting the corresponding genes in MCL [122].

### 3.4. Resistance to Temsirolimus and mTOR Inhibitors

As illustrated in Figure 2, mTOR is hyperactivated due to the constitutive activation of PI3K/AKT in MCL samples. In turn, the use of temsirolimus or everolimus was rapidly seen was a promising therapeutic option. The treatment of MCL cells with everolimus leads to a rapid dephosphorylation of mTOR and of two of its downstream targets, p70S6 kinase and eIF4E-binding protein (4E-BP1), both involved in the regulation of protein translation. However, in vitro experiments showed that, after a prolonged inactivation of mTOR by everolimus, AKT can be re-phosphorylated in a subset of cells, counteracting the effects of the drug [123]. The lack of everolimus activity is also linked to the recruitment of autophagy through an enhanced LC3 (microtubule-associated protein 1A/1B-light chain 3) activity and the accumulation of autophagosomes [122]. This observation raised the possibility that blocking autophagosome formation could restore everolimus-sensitivity. Temsirolimus, another mTOR inhibitor was shown to be active in R/R MCL patients, in particular in combination with ibrutinib treatment [80].

### 3.5. Resistance to BCL2-Targeting Agents

Among the different genetic lesions sustaining the tumorigenesis of MCL cells, abnormalities of different apoptosis signaling effectors have been documented. Among them, deletion of *BCL2L11* encoding the pro-apoptotic BIM protein, and amplification of the 18q21 locus leading to the overexpression of BCL-2, have been detected in MCL patients [59]. Importantly, homozygous deletion of BIM is mainly observed in MCL cell lines, and the loss of BIM protein found in about one third of MCL patients [124], is unlikely to be explained by the infrequent, heterozygous deletion of the gene reported by Tagawa and coworkers [125], and even not confirmed by others [126]. In this context, venetoclax (ABT-199), a BH3 mimetic with high specificity for BCL-2, has demonstrated notable activity as monotherapy in MCL patients [36]. By generating venetoclax-resistant cell lines, Tahir and coworkers described a variety of mechanisms conferring resistance, including the upregulation of MCL-1 or BCL-X_L_ anti-apoptotic proteins, and downregulation of the pro-apoptotic BIM and BAX [127].

Among the possible mechanism at the origin of BCL-2 overexpression, the E3 ubiquitin ligase, F-box only protein 10 (FBXO10), that targets BCL-2 for UPS-mediated degradation, is downregulated in MCL tumor cells. Thus, FBXO10 downregulation and BTK-mediated activation of the NF-κB canonical pathway may cooperate to sustain BCL-2 upregulation [128]. In turn, the targeting of BCL-2 in combination with pharmacological blockade of NF-κB has been tested in preclinical settings, showing synergy despite the onset of acquired resistance.

In parallel, by using MCL primary samples in vitro or engrafted in vivo (PDX), Zhao and coworkers reported that venetoclax drives the selection of clones having lost or a reduced copy number of the 18q21 amplicon that harbors BCL-2 [129]. Moreover, the reprogramming of super enhancer-driven transcription contributes to venetoclax resistance.

Finally, while analyzing the genetic determinants of the effectivity of the ibrutinib-venetoclax combination, Agarwal and colleagues found that all patients exhibiting alterations of *ATM* achieved a CR. By contrast, patients with deletion of the chromosome 9p21 that includes *CDKN2A/B* locus, and mutations in components of the SWI/SNF chromatin-remodeling complex, were resistant to or relapsed shortly after this therapy [130]. The analyses of circulating tumor DNA further showed that compromised SWI/SNF complex facilitated *BCL2L1* transcription and the upregulation of BCLX_L_.

## 4. Combination Therapies as Strategies to Overcome Drug Resistance

With the possible exception of ibrutinib, it seems unlikely that the biological drugs approved for the treatment of R/R patients will be used as single agents outside of maintenance strategies. They may rather have a role as part of combination therapy [131]. In this sense, several clinical trials are ongoing with different combinations such as temsirolimus plus rituximab (ORR 60%; 19% CR), bortezomib with R-HyperCVAD (95% CR) or lenalidomide plus rituximab (ORR 92%; 64% CR) [132,133]. Active ongoing trials combining new biological agents in R/R MCL patients are gathered in Table 2.

### 4.1. Targeting of Environmental Factors

#### 4.1.1. BCR Signaling

In order to increase the response to ibrutinib, this latest has been combined with rituximab, bendamustine, and R-CHOP in both untreated and refractory MCL cases [134,135,136]. In relapsed setting, ibrutinib/rituximab-based treatments resulted in higher responses, with ORR and CR of 88% and 44%, respectively. In combination with bendamustine and rituximab, the ORR was 94%, including 76% CR. Early phase study of the BTK inhibitor in combination with R-CHOP in previously untreated patients showed ORR of 94% with some manageable toxicity. Recently, combination of ibrutinib and venetoclax in R/R MCL showed ORR of 71% after 16 weeks of treatment. Most patients (67%) were negative for minimal residual disease (MRD) as assessed by flow cytometry [137].

#### 4.1.2. Adhesion Molecules

The capacity of MCL cells to reach and to colonize extranodal tissues is considered to depend on their transient interaction with vascular endothelium cells through adhesion molecules like selectins and integrins (“rolling”), and to migrate through the endothelium after chemokine receptor activation (“homing”), two processes that are conserved between most malignant B cells and their normal counterparts [138]. However, despite the pattern of early dissemination in MCL, only a few studies have investigated the expression and function of adhesion molecules and chemokine receptors related to these processes. Among them, high levels of functional C-X-C chemokine receptor type 4 (CXCR4) and 5 (CXCR5) and VLA-4 have been reported in MCL cell lines and primary cells. In agreement with an important role of these molecules in the migratory process of MCL cells and for MCL–stromal cell interactions and pseudo-emperipolesis [139], the CXCR4 antagonist plerixafor and the anti-VLA-4 antibody natalizumab have been shown to efficiently block CXCR4 and VLA-4 in in vitro and in vivo models of MCL, thus impeding physical interactions between MCL cells and MSCs, and rending these mobilized MCL cells more susceptible to standard therapies [140].

#### 4.1.3. IMiDs

While thalidomide has proved to be effective in R/R MCL patients as a single agent [141], in relapsed MCL patients, thalidomide-rituximab combination reached a ORR of 81% and a PFS of 20 months [142]. Another combination study of lenalidomide with rituximab has shown promising ORR in MCL patients with a poor response to initial treatment [143]. When used at frontline, beside the significant clinical benefit of this combination, a higher incidence of non-invasive skin and pancreatic cancers were reported [132]. At 5 years, 61% evaluable patients had remained in remission. Median PFS was not reached but estimated at 3 and 4 years, OS rates were 80.3% and 69.7%, respectively, thus confirming that combination therapy of lenalidomide with rituximab in first-line setting can result in long-term remission in MCL patients [144].

Regarding the combination of the IMiD with BTZ, preliminary results gathered in in vitro and in vivo models of BTZ-resistant MCL suggested that lenalidomide could partially overcome the resistance to the proteasome inhibitor mediated by the downregulation of IRF4 and MYC [111]. However, when lenalidomide and BTZ combination was administered to R/R patients for induction and maintenance therapy, outcomes were not satisfactory with median PFS and OS of 7 and 26 months, respectively, and ORR and CR of 39.6% and 15.1%, respectively [145]. These disappointing results were thought to be due to lenalidomide toxicity and inadequate dosing.

The efficacy of combining lenalidomide was also evaluated as upfront treatment in elderly patients (>70 years), with bendamustine and rituximab. After completion of induction therapy, 64% patients had CR and 36% were minimal residual disease (MRD) negative. Median PFS and OS were 42 and 53 months, respectively [146]. A major limitation of this combination was high incidence of serious infections, which makes this treatment probably inadequate for elderly patients.

### 4.2. New Therapeutics Antibodies

Despite the remarkable clinical efficacy of rituximab, a significant proportion of MCL patients experience disease relapse after a first step of clinical remission. To overcome resistance to rituximab-based regimens, a number of second generation anti-CD20 mAbs have been developed. Among these antibodies, ofatumumab is a fully human mAb that binds to an epitope encompassing both small and large loops of the extracellular domain of CD20. This binding epitope, distinct from that of rituximab, resides more proximal to the cell membrane. When compared with rituximab, ofatumumab exhibited enhanced CDC activity in a panel of MCL cell lines and prolonged survival in a mouse model of MCL. Importantly, significant activity of ofatumumab was observed in rituximab-resistant cases characterized by low levels of CD20 and/or high expression of complement inhibitory proteins [147]. However, first results with this Ab, in R/R MCL were disappointing in both single agent and combination settings [34].

Obinutuzumab (GA101), a type II glycol-engineered, humanized, anti-CD20 Ab was designed in an attempt to overcome common mechanisms of resistance to rituximab. To that aim, this Ab has non-fucosylated sugars on the Fc portion, associated with a more potent effector response, and has also the ability to cause homotypic adhesion, triggering a different mechanism of direct cell death (DCD) [148]. This antibody, previously approved in frontline and R/R follicular lymphoma and with known efficacy in preclinical models of MCL [149], has demonstrated its utility in clinical settings when combined with either venetoclax frontline therapy in untreated MCL (LYMA-1001 trial) [150], or in combination with ibrutinib and/or venetoclax in relapsed settings (OAsIs trial). In this second trial, both obinutuzumab-ibrutinib and obinutuzumab-ibrutinib-venetoclax combinations were well-tolerated and provided high disease control including CR at the molecular level (i.e., without detectable *CCND1* transcript) [151]. In the same line, obinutuzumab-venetoclax combination has also recently been suggested as a possible salvage therapy in nnMCL. Of special interest, this combination was well-tolerated and induced a CR after only two cycles and in a patient with prior refractoriness to bendamustine and ibrutinib [152].

A third new generation glycol-engineered anti-CD20 mAb, ublituximab (TG-1101) was engineered to have a low fucose content, conferring it enhanced ADCC activity when compared to rituximab, especially in rituximab-resistant cases with low expression of CD20. This Ab was well-tolerated and highly active in combination with ibrutinib in patients with R/R MCL [153]. In this phase 2 trial, among the 15 patients tested, a 87% ORR, including 33% CR, was reported, slightly superior to the 46% ORR and 17% CR observed in another cohort of R/R B-NHL when associating ublituximab to another inhibitor of PI3Kδ, umbralisib (U2 regimen) [154]. From a mechanistic point of view, first preclinical studies in MCL and other B-NHL cells co-cultured with stromal cells and macrophages, suggested that the U2 combination may cooperate with the blocking of the CD47 immune checkpoint by regulating genes related with cell architecture [154].

The membrane antigen CD74, that functions as a MHC class II chaperone, has been implicated in malignant B-cell growth and survival, making it a potential target for immunotherapy. The humanized version of the anti-CD74 mAb LL1, milatuzumab, exerts a direct tumoricidal effect in a disseminated mouse model of Burkitt lymphoma with a mechanism of action distinct from antibody-dependent cell-mediated cytotoxicity (ADCC) or CDC [155]. Of interest, therefore, it has been tested with success in preclinical model of MCL in combination with anti-CD20 mAb [156].

Bispecific T cell-engaging (BiTE) therapy consists of the transient engagement of CD3+ polyclonal T cells with malignant CD19+ B cells, resulting in T-cell-mediated, granzymes- and perforin-dependent lysis of tumor cells, and concomitant T-cell expansion and release of cytokines. In a phase 1 trial for heavily pretreated B-NHL patients including 24 MCL patients, the bispecific CD19/CD3 antibody blinatumomab showed a remarkable single agent activity with an ORR of 71% [157]. However, the shortness of the observed response was associated with the inability of blinatumomab to recruit competent cytotoxic T cells, leading to premature T-cell exhaustion. Thalidomide derivatives like lenalidomide have been shown to improve efficacy of anti-CD20 mAb (rituximab) through T and NK cell activation even in patients with previous relapse after rituximab-based therapy [143]. Based on this observation, the blinatumomab/lenalidomide combination regimen was evaluated in a phase 1 trial involving patients with R/R CD19+ B-NHL, including 3/18 MCL patients. At the median follow-up, combination-receiving patients achieved a 83% ORR, including 50% of CR, with a median PFS of 8.3 months, thus demonstrating the safety and the efficacy of this regimen in previously heavily treated patients [158].

### 4.3. Epigenetic Drugs

With the recent use of NGS technologies for the identification of lymphoma mutational landscape, it came out that different epigenetic deregulations were associated with B lymphomagenesis and lymphoma progression [159]. However, targeting epigenetic modification mechanisms is a relatively novel approach in MCL. Most of the preclinical studies have been centered on the evaluation of HDAC inhibitors, vorinostat being the main agent included in combination regimens with either proteasome, PI3K-AKT, CDK or BTK inhibitors [160]. From these different strategies, the combination of vorinostat plus BTZ has been evaluated in a phase 2 trial, but only a modest clinical activity was observed [161]. More recently, some bromodomain inhibitors with the capacity to targeting epigenetic readers of the BRD family, have been shown to synergistically induce apoptosis when combined to venetoclax, palbociclib, or panobinostat, in BTK wild-type, ibrutinib-resistant MCL cell lines characterized by the overexpression of antiapoptotic molecules like BCL-2, BCL-X_L_, XIAP, or with increased levels of CDK6 or AKT [162]. These promising results thus warrant the clinical evaluation of bromodomain inhibitors with other biological agents in R/R MCL patients.

Beside HDAC and bromodomain inhibition, cladribine, a hypomethylating agent that indirectly downregulates DNA methylation, has been used with vorinostat and rituximab in a phase 1/2 trial involving both naïve and R/R MCL patients. This triple combination reached an ORR of 97%, including 80% CR, with a 2-year PFS of 70.7% and OS of 86.9%, in previously untreated MCL patients. However, the ORR dropped to 39% in R/R MCL patients [163]. Improved results were reported in another trial using cladribine-rituximab combination in association with BTZ in B-NHL patients, including 24 MCL cases: the ORR and CR for both new and relapsed/refractory MCL cases were 85% and 77%, respectively [164]. Thus, although hypomethylating agent might show reduced single activity in R/R MCL patients, these last results warrant the evaluation of cladribine-rituximab backbone in further trials to determine whether the activity of this combination can be improved by the inclusion of additional biological agents.

### 4.4. Immune Checkpoint Inhibitors

Different immune checkpoint molecules expressed at the surface of tumor cells and accompanying immune cells, have been recently involved in the lowering of antitumor immunity. The main players are programmed death 1 (PD-1) and its ligands PD-L1 and PD-L2, cytotoxic T-lymphocyte activator 4 (CTLA-4), lymphocyte activation gene 3 (LAG-3), and CD200. MCL B cells were recently shown to express PD-1, PD-L1, and some degree of CD200 [165]. Accompanying T and NK cells (immune effectors), were positive for PD-1. PD-L1 expression on MCL cells inhibits T cell-mediated tumor cytotoxicity and their specific antitumor response. Indeed, in the presence of T cells, MCL cells increased their surface levels of PD-L1 in an IFNγ- and CD40-dependent manner. Despite initial in vitro and in vivo evidence for PD-L1 inhibition as a mechanism for effective enhancement of the T-cell response and T-cell-mediated killing of primary cells from PD-L1+ MCL patients, early clinical studies have not validated this approach as a successful strategy to treat patients with MCL [166]. However, as PD-L1 upregulation observed in MCL-T cell cocultures could be counteracted by either BTK or PI3K inhibition using ibrutinib or duvelisib [165], the combination of PD-1 blockade with BCR pathway inhibitors could represent a very promising combination [167].

Another immune checkpoint of interest in MCL is CD47. This myeloid checkpoint acts as a “don’t-eat-me” signal to macrophages and is found upregulated by tumor cells to evade the host’s immune response. ALX148 is a fusion protein comprised of a high affinity CD47 blocker linked to an inactive human Ig Fc region. This agent has been shown to increase the efficacy of the anti-CD20 mAbs rituximab and obinutuzumab in xenograft models of hematologic malignancies, bridging innate and adaptive immune responses including the activation of dendritic cells and a shift of tumor associated macrophages toward a pro-inflammatory phenotype [168]. In a phase 1 trial, a total of 20 CD20+ B-NHL patients, including four R/R MCL, received ALX148 in combination with rituximab. A 31% ORR was achieved in the most aggressive B-NHL cases, including DLBCL and MCL patients, with two MCL patients achieving a PR. ALX148 demonstrated excellent tolerability with favorable PK/PD characteristics and with unreached maximum tolerated dose [169].

As commented previously, T-cell exhaustion plays a major role in immune evasion in B-NHL, including MCL. CD27 is a co-stimulatory receptor involved in the negative regulation of T-cell activation following TCR engagement. Varlilumab (CDX-1127) is an agonistic IgG1 mAb that can bind CD27 and reverse the mechanism of T-cell exhaustion, allowing direct anti-tumoral activity in xenograft models of human lymphoma via ADCC [170]. Following previous phase 1 data supporting the safety and tolerability of single-agent varlilumab in advanced hematologic malignancies (ClinicalTrials.gov Identifier: NCT01460134), a randomized phase 2 study is currently ongoing, evaluating whether the varlimumab-mediated CD27 activation could synergize with the anti-PD-1 antibody, nivolumab, in R/R aggressive B-cell lymphomas, including MCL [171].

Another activation-induced costimulatory molecule with an important role in the regulation of immune responses is 4-1BB (CD137; TNFRS9). Validating the relevance of blocking 4-1BB or its natural ligand, 4-1BBL, in cancer, it has been shown that 4-1BB-mediated anti-cancer effects are based on its ability to facilitate the activation of cytotoxic T lymphocytes (CTL) and the production of IFN-γ. Accordingly, specific 4-1BB/CD137 agonistic antibodies can trigger costimulatory signals that enhance ADCC and elicit T-cell-mediated antitumor immune responses. A phase 1 study evaluated the activity of the anti-4-1BB antibody utomilumab in combination with rituximab in a total of 67 patients with CD20+ R/R B-NHL, including six MCL patients. The best overall response (BOR), including CR or PR, was observed in patients with MCL, FL, and DLBCL, with a favorable safety profile and clinical activity [172].

## 5. Conclusions

The main challenges in the management of MCL patients are tightly correlated with the biological diversity of the disease and its heterogeneous clinical presentation, that both underlie the existence of various morphological subtypes, distinct *IGHV* and *TP53* mutational status, together with the absence of actionable genetic variants that could define a common therapeutically amenable target for this disease. With standard chemotherapeutic regimens, physicians have to face frequent disease progression, recurrence, and limited disease-free interval. Among the above-mentioned therapies, intensified chemotherapy associated to rituximab or lenalidomide could allow better control of the disease in treatment-naïve patients; however those treatments that can converge to the blockade of BCR and NF-kB signal pathways (ibrutinib, everolimus) and/or to the impairment of apoptotic signaling (venetoclax), seemed to have the best therapeutic significance in the management of R/R patients. New biological agents or novel rationally based drug combinations including immunotherapeutic antibodies or CAR T-cell therapy will hopefully lead not only to better control of the disease, but also to the effective eradication of the residual clone. Finally, in an effort to achieve long-term remission without excessive toxicity, it is a safe bet that the development of genome-based precision medicine based on the last technological advances and on our growing knowledge on MCL biology will be the way to go.

## Figures and Tables

**Figure 1 cancers-12-01565-f001:**
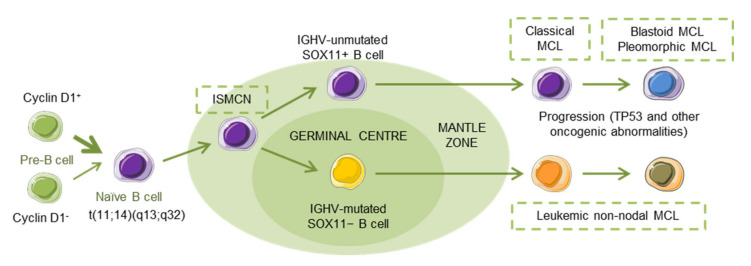
Hypothetical models of major mantle cell lymphoma (MCL) subtypes. Precursor B cells may colonize the inner portion of the mantle zone, representing in situ mantle cell neoplasia (ISMCN). After the introduction of additional genetic and molecular abnormalities, ISMCN may progress, involving or not the transit through the germinal center (GC), to classical MCL or leukemic non-nodal MCL, respectively. More frequently, classical MCL but also leukemic non-nodal MCL undergo additional molecular/cytogenetic abnormalities leading to clinical and sometimes to morphological progression. Adapted from Swerdlow et al. [6].

**Figure 2 cancers-12-01565-f002:**
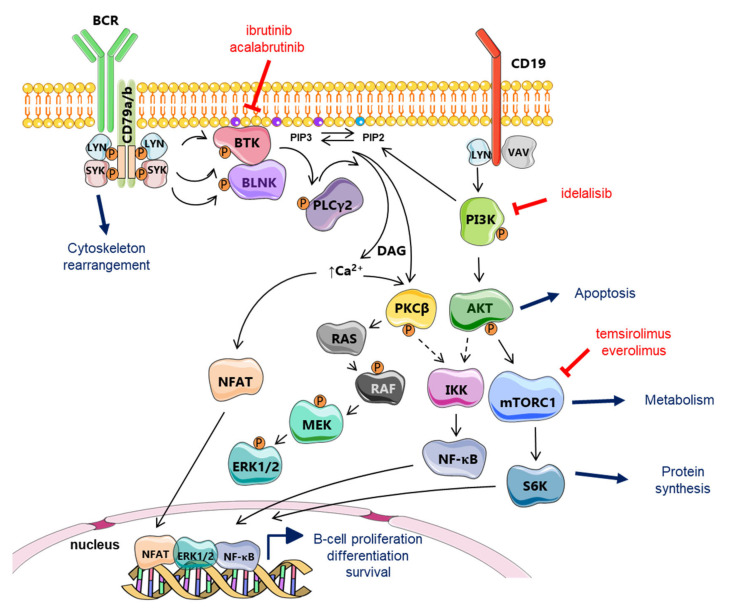
B-cell receptor signal transduction. After antigen ligation, LYN, SYK, and Bruton’s tyrosine kinase (BTK) are activated. B-cell adaptors such as B-cell linker (BLNK) fine-tune B-cell receptor (BCR) signals by efficiently connecting the kinases with the effectors. Activation of phospholipase (PL)Cγ2 leads to the release of intracellular Ca^2+^ and activation of protein kinase C (PKC); both of which are crucial for the activation of mitogen-activated protein kinase (MAPK), such as extracellular signal-regulated kinase (ERK) and transcription factors, including NF-κB and NFAT. BCR signaling can be efficiently targeted in MCL either by irreversible BTK inhibitors like the first-in-class ibrutinib and the second generation drug, acalabrutinib, or by means of the PI3Kδ inhibitor, idelalisib. Downstream mTOR kinase activity can be controlled by the mTORC1-targeting agents, everolimus and temsirolimus. Adapted from Herrera et al. [65].

**Figure 3 cancers-12-01565-f003:**
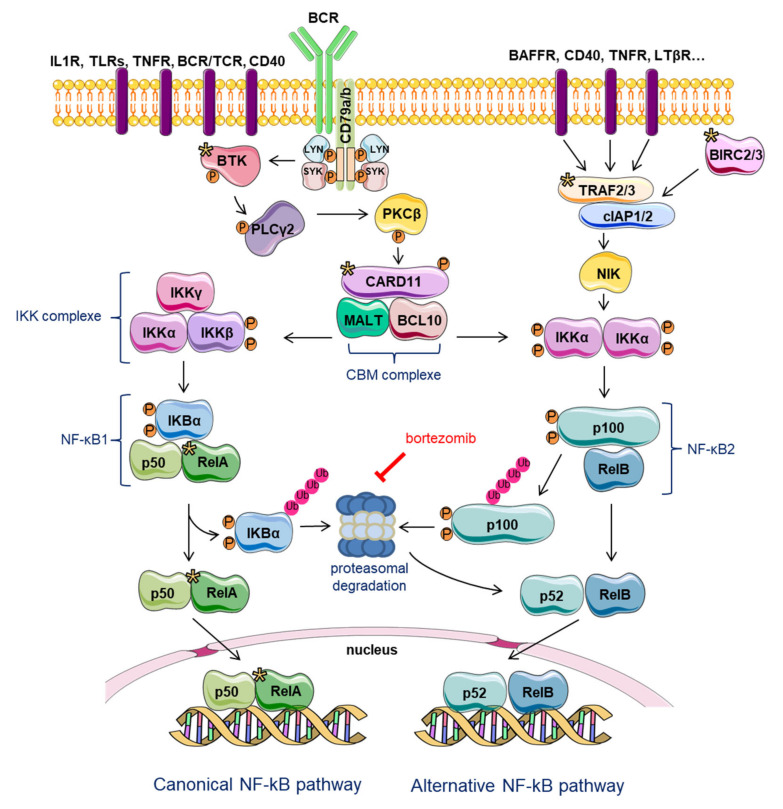
Canonical and alternative NF-κB signaling pathways. Canonical pathway is triggered by toll-like receptors (TLRs), pro-inflammatory cytokines such as IL1 and TNFα receptor, and CD40. It relies on inducible degradation of IκBs, particularly IκBα, leading to nuclear translocation of various NF-κB complexes, predominantly the p50/RelA dimer. Non-canonical NF-κB pathway relies on phosphorylation-induced p100 processing, which is triggered by signals from a subset of tumor necrosis factor α (TNFα) receptor (TNFR) members. This pathway is dependent on NF-κB-inducing kinase (NIK) and IκB kinase (IKK)α and mediates the activation of RelB/p52 complexes. Proteins potentially mutated in MCL are highlighted with a yellow star.

**Table 1 cancers-12-01565-t001:** Recurrent genomic alterations described for MCL patients and cell lines.

Gene	Frequency (Range) *	Protein Function	References
*ATM*	38–50%	DNA repair DNA damage response	[42,43,48,49,53,54]
*CCND1*	16–35%	Cell cycle regulation	[42,43,48,52,53,54]
*TP53*	14–31%	DNA damage response Cell cycle regulation	[42,43,48,49,53,54]
*MLL2*	14–20%	Epigenetic regulator (HMT)	[42,43]
*MLL3*	16%	Epigenetic regulator (HMT)	[42,54]
*WHSC1*	7–31%	Epigenetic regulator (HMT)	[42,43,54]
*BIRC3*	6–9%	Apoptosis regulator through TRAF2	[42,43,54]
*NOTCH1*	5–16%	NOTCH survival pathway	[42,43,52,53,54]
*NOTCH2*	5–6%	NOTCH survival pathway	[42,48]
*TRAF2*	7%	NF-κB pathway	[54]
*UBR5*	7–18%	Proteasome degradation(E3 ligase)	[42,53,54]
*RB1*	nd	Cell cycle regulation	[42]
*SMARCA4*	nd	Chromatin modifier	[42]
*CARD11*	5%	NF-κB pathway	[55]

Abbreviation: nd, not defined; HMT, histone methyl-transferase. * For the various cohorts, the range of percentages of genomic alterations is indicated with the corresponding references. Somatic mutations lead to the constitutive activation of signaling pathway downstream of the mutated protein. Point mutation of *BTK* (*BTK*^C481S^) leads to the chronic activation of BCR/NF-κB signaling and AKT pathway [56,57]. *BIRC3* and *TRAF2* mutations as well as *CARD11* mutation are associated with the chronic activation of the canonical NF-κB pathway [55]. *NOTCH1* mutations are located in an exon coding for the PEST sequence, causing truncation of the C-terminal part of the protein, and chronic activation of the NOTCH pathway [52]. *ATM* mutation sustains defects of DNA repair machinery and impairs apoptosis [58]. p53 and ATM are interrelated, both being involved in the sensing of DNA damage and in the balance cell cycle/apoptosis. In agreement, MCL tumor cells display a high chromosome instability and numerous chromosome alterations [59].

**Table 2 cancers-12-01565-t002:** Active clinical trials combining new biological agents in R/R MCL.

Drug Combination	Targets	Study Number	Efficiency
Obinutuzumab + Venetoclax + Ibrutinib	CD20, BCL2, BTK	NCT02558816	No results available
Ibrutinib + Lenalidomide + Rituximab	BTK, CRBN/CD20	NCT02446236	No results available
Alisertib + Bortezomib + Rituximab	Aurora A kinase, 20S proteasome, CD20	NCT01695941	No results available
Ibrutinib + Bortezomib	BTK, 20S proteasome	NCT02356458	No results available
Rituximab + Bendamustine + Ibrutinib	CD20, alkylating agent, BTK	NCT01479842	No results available
Lenalidomide + Ibrutinib	CRBN, BTK	NCT01955499	No results available
BKM120 + Rituximab	PI3K, CD20	NCT02049541	No results available
Entospletinib + Obinutuzumab	SYK, CD20	NCT03010358	No results available
Everolimus + Lenalidomide	mTOR, CRBN	NCT01075321	9.8% CR, 19.5% PR, 39% SD, 29.3% progression

Abbreviations: complete response (CR), partial response (PR), stable disease (SD).

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
