# Peer review of "Management of Drug Resistance in Mantle Cell Lymphoma"

_cancers, 2020, doi:10.3390/cancers12061565_

Round 1

Reviewer 1 Report

In this manuscript, the authors focused on Mantle cell lymphoma (MCL) and reviewed the physiopathology, subtypes, biological features, prognostic factors, and therapy, especially the management of drug resistance in MCL. A wide range of topics focusing on the relationship between molecular signatures and drug resistance to various types of treatments, were moderately covered, respectively. Overall, since this manuscript is well written and has clinically-significant interests in MCL treatment, I believe that, with minor revision, this manuscript should be published in Cancers.

In this manuscript, the author mainly described the efficacy of the different targeted therapies including proteasome inhibitors, immunomodulatory drugs, and tyrosine kinase inhibitors. Meanwhile, in addition to the drug resistance, it is also documented that the critical side effects of those anti-cancer drugs are also big issues in clinical (eg. bortezomib:

nausea & vomiting, constipation etc…). To provide a better quality of manuscript and strengthen their review, ideally, the side effects by various types of treatments should be discussed and described.

Author Response

Reviewer 1

In this manuscript, the authors focused on Mantle cell lymphoma (MCL) and reviewed the physiopathology, subtypes, biological features, prognostic factors, and therapy, especially the management of drug resistance in MCL. A wide range of topics focusing on the relationship between molecular signatures and drug resistance to various types of treatments, were moderately covered, respectively. Overall, since this manuscript is well written and has clinically-significant interests in MCL treatment, I believe that, with minor revision, this manuscript should be published in Cancers.

In this manuscript, the author mainly described the efficacy of the different targeted therapies including proteasome inhibitors, immunomodulatory drugs, and tyrosine kinase inhibitors. Meanwhile, in addition to the drug resistance, it is also documented that the critical side effects of those anti-cancer drugs are also big issues in clinical (eg. bortezomib: nausea & vomiting, constipation etc…). To provide a better quality of manuscript and strengthen their review, ideally, the side effects by various types of treatments should be discussed and described.

Answer: we thank the reviewer for his/her positive evaluation of our work. In agreement with his/her suggestion, we rewrote the chapter 1.3. focused on MCL therapy and included in this new version, a more complete description of current biological agents together with their main side-effects (page 4, lines 142-158).

Reviewer 2 Report

This is a very nicely written and impressively comprehensive review on MCL resistance. To put the topic into context, the authors start with a concise review of MCL biology, followed by an overview of the molecular genetics of MCL. The main topic of resistance in MCL is followed by a discussion of the various targeted therapy and immunotherapy approaches.

The review was easy to follow and will be of interest for a broad readership.

I have only some minor comments:

line 27: through, not though

line 32: abbreviation NHL is never spelled out

line 34: it seems strange to have an age range (29-85) in such a general introduction to the topic. Are the authors referring to a population in a particular clinical trial? Or in other words, does MCL never occur in patients <29 years of age.

line 162: ATM = ataxia telangectasia-mutated (mutated was missing)

Table 1: show range of % genomic alterations to give better overview, e.g. 38-50% for ATM

line 309-311: I find the term de novo resistance a bit misleading. A common way to denominate resistance mechanisms (widely used for BCR-ABL inhibition) is primary vs. secondary resistance, as also the term acquired resistance does not describe the process properly. Often it is not the acquisition, but rather the (positive) selection of a particular protein/pathways/cell/clonal population that leads to resistance.

Author Response

Reviewer 2

This is a very nicely written and impressively comprehensive review on MCL resistance. To put the topic into context, the authors start with a concise review of MCL biology, followed by an overview of the molecular genetics of MCL. The main topic of resistance in MCL is followed by a discussion of the various targeted therapy and immunotherapy approaches.

The review was easy to follow and will be of interest for a broad readership.

I have only some minor comments:

line 27: through, not though

line 32: abbreviation NHL is never spelled out

line 34: it seems strange to have an age range (29-85) in such a general introduction to the topic. Are the authors referring to a population in a particular clinical trial? Or in other words, does MCL never occur in patients <29 years of age.

line 162: ATM = ataxia telangectasia-mutated (mutated was missing)

Table 1: show range of % genomic alterations to give better overview, e.g. 38-50% for ATM

Answer: we thank the reviewer 2 for his/her positive evaluation of our work and for his/her comments. We apologize for the typographical errors left on the manuscript. We corrected all of them. We modified the Table 1 and showed the range of percentage of genomic alterations as required (page 6).

line 309-311: I find the term de novo resistance a bit misleading. A common way to denominate resistance mechanisms (widely used for BCR-ABL inhibition) is primary vs. secondary resistance, as also the term acquired resistance does not describe the process properly. Often it is not the acquisition, but rather the (positive) selection of a particular protein/pathways/cell/clonal population that leads to resistance.

Answer: we agree with the reviewer’s comment. De novo resistance is synonymous with intrinsec or primary resistance and is defined as no response to a drug for a patient who received no previous treatment. By contrast, acquired or secondary resistance arises for patients receiving a treatment after a more or less long period of time. De novo and acquired resistance are jargon terms but they are generally accepted and used in the medical literature including scientific papers related to MCL (e.g. Luanpitpong et al, Cancers 2019, 11, 576; Zhao et al, Nat Commun 2017, 8, 14920). We included the definition of these terms in the revised version of the manuscript (page 11, lines 350 and 352).

Reviewer 3 Report

General comments:

This is a review on drug-resistance management in MCL, but the first 10 pages (i.e. the first half of the review) are devoted purely to recapitulation of the pathophysiology and therapy of MCL. Mechanisms of resistance are described from page 11 (to page 20). Some chapters, however, do not discuss mechanisms of drug resistance or management of R/R MCL, but merely recapitulate results of available data from completed or currently evaluated clinical trials (e.g. 4.3. hypomethylating agents, large part of 4.2. New therapeutic antibodies). Such chapters should be re-written to comply with the focus of the review that the readers would suggest from the title. What is the role of hypomethylating agents in the management of R/R MCL? Why new generation anti-CD20 antibodies work better than rituximab? Do they work in CD20-negative patients?

In addition, the current version contains several misleading facts about MCL and its therapy in the first half of the review.

Last but not least, the English language editing is necessary, as many phrases are cumbersome or inappropriate.

Specific comments:

Line 34: “a median age of approximately 60 years (range 29-85).

-median age at diagnosis in MCL is approx. 67 years (this is a disease of the elderly)

-what is meant by “range 29-85”M?

Line 34: “, and a high male-to-female ratio (around 2-7:1)

-in the real-world cohorts of patients the ratio is 2-3:1 (not 7:1).

Line 37-38: “Other extranodal sites are frequently involved, including the GIT tract and Waldeyer´s ring.”

-Waldeyer´s ring is not considered “extra-nodal” site.

Line 38-39: The clinical evolution is usually very aggressive, with poor responses to treatment and frequent relapses.

-overall response rate in real-world cohorts of patients is approx. 70-80% (up to 90% in some clinical studies) and median PFS in the elderly patients after R-CHOP-based induction and rituximab maintenance is approx. 5 years. I would not describe this as “poor” response to treatment.

Line 57: “the classic, the small cell, the blastoid and the pleomorphic.”

In the daily clinical practice and in most published reports, only 3 relevant categories are usually used: classic, pleomorphic and blastic/blastoid.

Line 95: Although there is no accepted standard, there are two main approaches: HyperCVAD……”

-first, there are of course accepted standards of care

-second, these standards of care, however, do not include HyperCVAD1. Although R-HyperCVAD alternating with high-dose MTX/araC was active in a single-center experience, it has not performed as well in cooperative group settings2-4. In addition, the regimen is toxic1. Several intensive induction regimens implementing rituximab and high-dose araC (HDAC) have been used in Europe: the “Nordic MCL2” protocol (R-Maxi-CHOP / R-HDAC, 3+3 cycles), R-CHOP / R-DHAP (3+3 cycles), and R-DHAP (4 cycles) belong to most commonly used (reviewed in Spurgeon et al)5.

-third, what was the “second” main approach the authors suggested at line 95?

Line 104: “…fludarabine and cyclophosphamide (FC) … have been the mainstay of therapy for the elderly MCL.”

-first, it was demonstrated already in the landmark study of European MCL Network that R-FC + rituximab maintenance is inferior to R-CHOP + rituximab maintenance6. R-FC thus cannot be considered an acceptable “standard of care”.

-second, bortezomib + R-CHP (VR-CAP) regime has demonstrated survival benefit over R-CHOP in a phase 3 trial and should be mentioned among the standards of care7

-third, rituximab is considered a standard of care for all newly diagnosed MCL patients and this must be mentioned

Line 105: “Maintenance therapy is a well-established approach…”

-must be corrected to “Maintenance therapy with rituximab after R-CHOP-based induction is a well-established approach…”, because other forms of maintenance therapy (lenalidomide, bortezomib) have not demonstrated benefit or are being currently tested (ibrutinib). Rituximab maintenance has demonstrated clear survival benefit only after R-CHOP-based induction.

Line 109: “For relapsed/refractory (R/R) patients, autologous stem cell transplant (ASCT) is generally used,…”

-first, ASCT is generally used in the front-line setting, where it is still considered a standard of care for younger/fit patients. Consequently, it is almost never used in the real-cohort patients in R/R MCL.

-second, allogeneic stem cell transplantation should be mentioned as an alternative, especially in younger patients with early relapse or MCL refractory to induction therapy. As it is probably the only modality that can induce long-term control and (potentially) cure of R/R MCL. For CAR T-cell-based approaches, the data are still immature.

Line 119: “… the development of CAR T-cells (citation= Kochenderfer from 2009).”

-in addition to a citation from 2009, the authors must cite the results of a phase 2 trial recently published in NEJM? Wang M et al. KTE-X19 CAR T-Cell Therapy in Relapsed or Refractory Mantle-Cell Lymphoma. Engl J Med. 2020 Apr 2;382(14):1331-1342. doi: 10.1056/NEJMoa1914347.

Line 170: “MML3”

-should be “MLL3”

Line 323-324: “However, a longer follow-up is needed to conclude on acalabrutinib efficacy and lack of innate/acquired resistance.”

-“lack of innate/acquired resistance” should be omitted or re-phrased. Acalabrutinib belongs to 2nd generation BTK inhibitors that have been designed to have less off-target effects than ibrutinib due to more specific binding to BTK. Mode-of-action of acalabrutinib, however, is the same as that of ibrutinib, i.e. irreversible inhibition of BTK on cysteine residue. Consequently, the same mechanisms of resistance can be expected in acalabrutinib-resistant / refractory patients. Potential lack of innate/acquired resistance just due to more specific binding to BTK is a misleading speculation.

Line 356: “These data question the use of the ibrutinib and idelalisib for the treatment of MCL patietns for long periods”.

-based on one preclinical study the judgement should be less categorical and should mention only ibrutinib, as idelalisib is currently not (and will never be) used in the therapy of MCL.

Line 325: “Intrinsec”. Should be “Intrinsic”.

Line 361: “By modifying the cell cycle, CCND1 mutations contribute to ibrutinib resistance.”

-please, add citation(s) to support and explain this statement.

Line 369: “The upregulation of MYC and mTORC1 underlies OXPHOS…”

-please re-phrase, so that the reader understands, what is the meaning  of the sentence.

Line 378: “Over the past, proteasome inhibition has been demonstrated….”

-should be “Over the past years, proteasome inhibition has been demonstrated…”.

Lines 408-409: MCL-1 interacts with and blocks NOXA, counteracting the ignition of the apoptotic cascade mediated by this BH3-only protein.

-please, re-write. NOXA is a pro-apoptotic sensitizer, not an activator or effector pro-apoptotic protein. By definition, it cannot elicit apoptosis8. Rather, binding of NOXA to MCL1 might trigger apoptosis only in case of displacement of an activator or effector pro-apoptotic player, e.g. BIM, BAX or BAK1 bound to MCL1.

Lines 423-451: Resistance to lenalidomide.

Lenalidomide mode-of-action in MCL is mediated predominantly through NK Cell-Mediated Cytotoxicity9, 10. The authors should at least comment on that. Mechanisms of resistance to NK-cell-based immunotherapy are nicely reviewed e.g. in the paper of Sordo-Bahamonde et al11.

Line 455: “Deletion of BCL2L11 encoding the pro-apoptotic BIM protein…”.

The data on deletions of BIM in MCL remain controversial. Tagawa et al reported only heterozygous (not homozygous) deletions of BIM in 5 of 27 (18.5%) MCL patients12. Mestre-Escorihuela and colleagues focused mainly on analysis of cell lines and the only information provided about BIM status in primary MCL samples was based on IHC analysis by tissue microarray with loss of BIM protein expression detected in 7 of 22 (33%) patient samples13. Prukova et al analyzed 24 high-risk MIPI patients by aCGH and found only one patient with monoallelic BIM deletion in the context of entire loss of long arms of chromosome 2 within a complex karyotype14. Katz et al focused on genetic proof of concept of biallelic BIM gene deletion during MCL lymphomagenesis, however, did not analyze primary MCL cells15.

Page 591: “vorinostat-BTZ combo has been entered in phase 2 trial”

-please, rephrase.

Page 593: “Associations with other epigenetic drugs have also been evaluated.”

-please, rephrase… Efficacy of other epigenetic drugs has been evaluated?

  1. Chen, R. W.; Li, H.;  Bernstein, S. H.;  Kahwash, S.;  Rimsza, L. M.;  Forman, S. J.;  Constine, L.;  Shea, T. C.;  Cashen, A. F.;  Blum, K. A.;  Fenske, T. S.;  Barr, P. M.;  Phillips, T.;  Leblanc, M.;  Fisher, R. I.;  Cheson, B. D.;  Smith, S. M.;  Faham, M.;  Wilkins, J.;  Leonard, J. P.;  Kahl, B. S.; Friedberg, J. W., RB but not R-HCVAD is a feasible induction regimen prior to auto-HCT in frontline MCL: results of SWOG Study S1106. British journal of haematology 2017, 176 (5), 759-769.
  2. Chihara, D.; Cheah, C. Y.;  Westin, J. R.;  Fayad, L. E.;  Rodriguez, M. A.;  Hagemeister, F. B.;  Pro, B.;  McLaughlin, P.;  Younes, A.;  Samaniego, F.;  Goy, A.;  Cabanillas, F.;  Kantarjian, H.;  Kwak, L. W.;  Wang, M. L.; Romaguera, J. E., Rituximab plus hyper-CVAD alternating with MTX/Ara-C in patients with newly diagnosed mantle cell lymphoma: 15-year follow-up of a phase II study from the MD Anderson Cancer Center. British journal of haematology 2016, 172 (1), 80-8.
  3. Bernstein, S. H.; Epner, E.;  Unger, J. M.;  Leblanc, M.;  Cebula, E.;  Burack, R.;  Rimsza, L.;  Miller, T. P.; Fisher, R. I., A phase II multicenter trial of hyperCVAD MTX/Ara-C and rituximab in patients with previously untreated mantle cell lymphoma; SWOG 0213. Annals of oncology : official journal of the European Society for Medical Oncology 2013, 24 (6), 1587-93.
  4. Merli, F.; Luminari, S.;  Ilariucci, F.;  Petrini, M.;  Visco, C.;  Ambrosetti, A.;  Stelitano, C.;  Caracciolo, F.;  Di Renzo, N.;  Angrilli, F.;  Carella, A. M.;  Capodanno, I.;  Barbolini, E.;  Galimberti, S.; Federico, M., Rituximab plus HyperCVAD alternating with high dose cytarabine and methotrexate for the initial treatment of patients with mantle cell lymphoma, a multicentre trial from Gruppo Italiano Studio Linfomi. British journal of haematology 2012, 156 (3), 346-53.
  5. Spurgeon, S. E.; Till, B. G.;  Martin, P.;  Goy, A. H.;  Dreyling, M. P.;  Gopal, A. K.;  LeBlanc, M.;  Leonard, J. P.;  Friedberg, J. W.;  Baizer, L.;  Little, R. F.;  Kahl, B. S.; Smith, M. R., Recommendations for Clinical Trial Development in Mantle Cell Lymphoma. Journal of the National Cancer Institute 2017, 109 (1).
  6. Kluin-Nelemans, H. C.; Hoster, E.;  Hermine, O.;  Walewski, J.;  Trneny, M.;  Geisler, C. H.;  Stilgenbauer, S.;  Thieblemont, C.;  Vehling-Kaiser, U.;  Doorduijn, J. K.;  Coiffier, B.;  Forstpointner, R.;  Tilly, H.;  Kanz, L.;  Feugier, P.;  Szymczyk, M.;  Hallek, M.;  Kremers, S.;  Lepeu, G.;  Sanhes, L.;  Zijlstra, J. M.;  Bouabdallah, R.;  Lugtenburg, P. J.;  Macro, M.;  Pfreundschuh, M.;  Prochazka, V.;  Di Raimondo, F.;  Ribrag, V.;  Uppenkamp, M.;  Andre, M.;  Klapper, W.;  Hiddemann, W.;  Unterhalt, M.; Dreyling, M. H., Treatment of older patients with mantle-cell lymphoma. The New England journal of medicine 2012, 367 (6), 520-31.
  7. Robak, T.; Jin, J.;  Pylypenko, H.;  Verhoef, G.;  Siritanaratkul, N.;  Drach, J.;  Raderer, M.;  Mayer, J.;  Pereira, J.;  Tumyan, G.;  Okamoto, R.;  Nakahara, S.;  Hu, P.;  Appiani, C.;  Nemat, S.; Cavalli, F., Frontline bortezomib, rituximab, cyclophosphamide, doxorubicin, and prednisone (VR-CAP) versus rituximab, cyclophosphamide, doxorubicin, vincristine, and prednisone (R-CHOP) in transplantation-ineligible patients with newly diagnosed mantle cell lymphoma: final overall survival results of a randomised, open-label, phase 3 study. The Lancet. Oncology 2018, 19 (11), 1449-1458.
  8. Montero, J.; Letai, A., Why do BCL-2 inhibitors work and where should we use them in the clinic? Cell death and differentiation 2018, 25 (1), 56-64.
  9. Hagner, P. R.; Chiu, H.;  Ortiz, M.;  Apollonio, B.;  Wang, M.;  Couto, S.;  Waldman, M. F.;  Flynt, E.;  Ramsay, A. G.;  Trotter, M.;  Gandhi, A. K.;  Chopra, R.; Thakurta, A., Activity of lenalidomide in mantle cell lymphoma can be explained by NK cell-mediated cytotoxicity. British journal of haematology 2017, 179 (3), 399-409.
  10. Gribben, J. G.; Fowler, N.; Morschhauser, F., Mechanisms of Action of Lenalidomide in B-Cell Non-Hodgkin Lymphoma. Journal of clinical oncology : official journal of the American Society of Clinical Oncology 2015, 33 (25), 2803-11.
  11. Sordo-Bahamonde, C.; Vitale, M.;  Lorenzo-Herrero, S.;  López-Soto, A.; Gonzalez, S., Mechanisms of Resistance to NK Cell Immunotherapy. Cancers 2020, 12 (4).
  12. Tagawa, H.; Karnan, S.;  Suzuki, R.;  Matsuo, K.;  Zhang, X.;  Ota, A.;  Morishima, Y.;  Nakamura, S.; Seto, M., Genome-wide array-based CGH for mantle cell lymphoma: identification of homozygous deletions of the proapoptotic gene BIM. Oncogene 2005, 24 (8), 1348-58.
  13. Mestre-Escorihuela, C.; Rubio-Moscardo, F.;  Richter, J. A.;  Siebert, R.;  Climent, J.;  Fresquet, V.;  Beltran, E.;  Agirre, X.;  Marugan, I.;  Marin, M.;  Rosenwald, A.;  Sugimoto, K. J.;  Wheat, L. M.;  Karran, E. L.;  Garcia, J. F.;  Sanchez, L.;  Prosper, F.;  Staudt, L. M.;  Pinkel, D.;  Dyer, M. J.; Martinez-Climent, J. A., Homozygous deletions localize novel tumor suppressor genes in B-cell lymphomas. Blood 2007, 109 (1), 271-80.
  14. Prukova, D.; Andera, L.;  Nahacka, Z.;  Karolova, J.;  Svaton, M.;  Klanova, M.;  Havranek, O.;  Soukup, J.;  Svobodova, K.;  Zemanova, Z.;  Tuskova, D.;  Pokorna, E.;  Helman, K.;  Forsterova, K.;  Pacheco-Blanco, M.;  Vockova, P.;  Berkova, A.;  Fronkova, E.;  Trneny, M.; Klener, P., Co-targeting of BCL2 with venetoclax and MCL1 with S63845 is synthetically lethal in vivo in relapsed mantle cell lymphoma. Clinical cancer research : an official journal of the American Association for Cancer Research 2019.
  15. Katz, S. G.; Labelle, J. L.;  Meng, H.;  Valeriano, R. P.;  Fisher, J. K.;  Sun, H.;  Rodig, S. J.;  Kleinstein, S. H.; Walensky, L. D., Mantle cell lymphoma in cyclin D1 transgenic mice with Bim-deficient B cells. Blood 2014, 123 (6), 884-93.

Author Response

This is a review on drug-resistance management in MCL, but the first 10 pages (i.e. the first half of the review) are devoted purely to recapitulation of the pathophysiology and therapy of MCL.

Answer: since this review is destinated to a broad range of readers who are not necessarily familiar with aggressive B-cell lymphoma, we believe that the description of the disease, including genomic abnormalities that sustain abnormal cell signalization and rationally-based current targeted therapies, is necessary. To our understanding, this claim is supported by the first comment of Reviewer 2. In addition, following reviewer 1’s request of a better description of the current treatments, including side-effects, we rewrote the chapter 1.3 and extended the description of adverse events associated to the main targeted therapies used in MCL (page 3, lines 142-158). The chapter 2 centered on MCL genomic abnormalities was shortened partly upon the removal of the chapter titled “TNFR, CD40 and BAFFR signaling”, which has been replaced by the chapter 2.4.4. “PI3K/AKT/mTOR signaling pathway” (page 10, lines 337-345).

Mechanisms of resistance are described from page 11 (to page 20). Some chapters, however, do not discuss mechanisms of drug resistance or management of R/R MCL, but merely recapitulate results of available data from completed or currently evaluated clinical trials (e.g. 4.3. hypomethylating agents, large part of 4.2. New therapeutic antibodies). Such chapters should be re-written to comply with the focus of the review that the readers would suggest from the title. What is the role of hypomethylating agents in the management of R/R MCL? Why new generation anti-CD20 antibodies work better than rituximab? Do they work in CD20-negative patients?

Answer: following reviewer’s request, we rewrote the chapter 4 with additional emphasis on resistance mechanisms. In particular, within the chapter 4.2 we described more appropriately the advantages of new anti-CD20 mAbs over rituximab and how their use can bypass rituximab refractoriness (page 17, lines 599-611).

In addition, the current version contains several misleading facts about MCL and its therapy in the first half of the review.

Answer: we apologize for these errors and corrected them carefully (see the specific comment section).

Last but not least, the English language editing is necessary, as many phrases are cumbersome or inappropriate.

Answer: our manuscript has been carefully corrected for English language and checked by a fluent English speaker.

Specific comments:

Line 34: “a median age of approximately 60 years (range 29-85).

-median age at diagnosis in MCL is approx. 67 years (this is a disease of the elderly)

-what is meant by “range 29-85”M?

Line 34: “, and a high male-to-female ratio (around 2-7:1)

-in the real-world cohorts of patients the ratio is 2-3:1 (not 7:1).

Answer: we corrected the median age at diagnosis and the male/female ratio according to the comment of the reviewer, and delete the range 29-85 which was confusing (page 1, lines 34-35).

Line 37-38: “Other extranodal sites are frequently involved, including the GIT tract and Waldeyer´s ring.”

-Waldeyer´s ring is not considered “extra-nodal” site.

Answer: in agreement with this comment, this sentence was modified as follows: “Waldeyer’s ring and extranodal sites including the gastrointestinal tract, are also frequently involved “ (page 1, lines 37- 38).

Line 38-39: The clinical evolution is usually very aggressive, with poor responses to treatment and frequent relapses.

-overall response rate in real-world cohorts of patients is approx. 70-80% (up to 90% in some clinical studies) and median PFS in the elderly patients after R-CHOP-based induction and rituximab maintenance is approx. 5 years. I would not describe this as “poor” response to treatment.

Answer: we agree with this comment. We modified the sentence by the following text in the new version of the manuscript : “The clinical evolution is usually very aggressive, and despite overall response rates above 70% with standard immunochemotherapeutic schemes (see below), few patients can be cured (page 1, lines 38-40).

Line 57: “the classic, the small cell, the blastoid and the pleomorphic.”

In the daily clinical practice and in most published reports, only 3 relevant categories are usually used: classic, pleomorphic and blastic/blastoid.

Answer: we agreed with this point and modified the text on MCL subtypes description. The corresponding sentence is now: “Morphologically, three main subtypes of MCL are recognized: the classic, the blastic/blastoid and the pleomorphic variants” (page 2, lines 57-58).

Line 95: Although there is no accepted standard, there are two main approaches: HyperCVAD……”

-first, there are of course accepted standards of care

-second, these standards of care, however, do not include HyperCVAD1. Although R-HyperCVAD alternating with high-dose MTX/araC was active in a single-center experience, it has not performed as well in cooperative group settings2-4. In addition, the regimen is toxic1. Several intensive induction regimens implementing rituximab and high-dose araC (HDAC) have been used in Europe: the “Nordic MCL2” protocol (R-Maxi-CHOP / R-HDAC, 3+3 cycles), R-CHOP / R-DHAP (3+3 cycles), and R-DHAP (4 cycles) belong to most commonly used (reviewed in Spurgeon et al)5.

-third, what was the “second” main approach the authors suggested at line 95?

Line 104: “…fludarabine and cyclophosphamide (FC) … have been the mainstay of therapy for the elderly MCL.”

-first, it was demonstrated already in the landmark study of European MCL Network that R-FC + rituximab maintenance is inferior to R-CHOP + rituximab maintenance6. R-FC thus cannot be considered an acceptable “standard of care”.

-second, bortezomib + R-CHP (VR-CAP) regime has demonstrated survival benefit over R-CHOP in a phase 3 trial and should be mentioned among the standards of care7

-third, rituximab is considered a standard of care for all newly diagnosed MCL patients and this must be mentioned

Line 105: “Maintenance therapy is a well-established approach…”

-must be corrected to “Maintenance therapy with rituximab after R-CHOP-based induction is a well-established approach…”, because other forms of maintenance therapy (lenalidomide, bortezomib) have not demonstrated benefit or are being currently tested (ibrutinib). Rituximab maintenance has demonstrated clear survival benefit only after R-CHOP-based induction.

Line 109: “For relapsed/refractory (R/R) patients, autologous stem cell transplant (ASCT) is generally used,…”

-first, ASCT is generally used in the front-line setting, where it is still considered a standard of care for younger/fit patients. Consequently, it is almost never used in the real-cohort patients in R/R MCL.

-second, allogeneic stem cell transplantation should be mentioned as an alternative, especially in younger patients with early relapse or MCL refractory to induction therapy. As it is probably the only modality that can induce long-term control and (potentially) cure of R/R MCL. For CAR T-cell-based approaches, the data are still immature.

Line 119: “… the development of CAR T-cells (citation= Kochenderfer from 2009).”

-in addition to a citation from 2009, the authors must cite the results of a phase 2 trial recently published in NEJM? Wang M et al. KTE-X19 CAR T-Cell Therapy in Relapsed or Refractory Mantle-Cell Lymphoma. Engl J Med. 2020 Apr 2;382(14):1331-1342. doi: 10.1056/NEJMoa1914347.

Answer: to be consistent with reviewer’s comments and criticisms, the entire chapter 1.3 has been rewritten and now includes most of the references suggested by the reviewer (pages 3-5, lines 91-176). We gratefully acknowledge the suggestion of the reviewer, which permitted us to consistently improve the meaningfulness of this chapter.

Line 170: “MML3”

-should be “MLL3”

Answer: this typographical error has been corrected.

Line 323-324: “However, a longer follow-up is needed to conclude on acalabrutinib efficacy and lack of innate/acquired resistance.”

-“lack of innate/acquired resistance” should be omitted or re-phrased. Acalabrutinib belongs to 2nd generation BTK inhibitors that have been designed to have less off-target effects than ibrutinib due to more specific binding to BTK. Mode-of-action of acalabrutinib, however, is the same as that of ibrutinib, i.e. irreversible inhibition of BTK on cysteine residue. Consequently, the same mechanisms of resistance can be expected in acalabrutinib-resistant / refractory patients. Potential lack of innate/acquired resistance just due to more specific binding to BTK is a misleading speculation.

Answer: we agree with this comment. The term “lack of innate/acquired resistance” has been omitted in the new version of the manuscript (page 11, lines 364-366). 

Line 356: “These data question the use of the ibrutinib and idelalisib for the treatment of MCL patietns for long periods”.

-based on one preclinical study the judgement should be less categorical and should mention only ibrutinib, as idelalisib is currently not (and will never be) used in the therapy of MCL.

Answer: we agree with this comment. The reference to idelalisib was removed and the sentence was changed to “Given that ibrutinib is currently being used for the treatment of R/R MCL patients, these data may question its administration for long periods” (page 12, lines 398-400). 

Line 325: “Intrinsec”. Should be “Intrinsic”.

Answer: this typographical error has been corrected.

Line 361: “By modifying the cell cycle, CCND1 mutations contribute to ibrutinib resistance.”

-please, add citation(s) to support and explain this statement.

Answer: Mohanty and coworkers described CCND1 mutations that affect the degradation rate of cyclin D1 by interfering with the ubiquitin/proteasome system (ref. 46 cited in the text: Mohanty et al, Oncotarget 2016,7, 73558). In turn, cyclin D1 accumulates in the nucleus of tumor cells. It is well-known that the nuclear form of cyclin D1 is oncogenic, deregulating the cell cycle and increasing cell proliferation (for reviews see Hydbring et al, Nat Rev Mol Cell Biol, 2016, 17, 280; Tchakarska and Sola, Cell Cycle, 2020, 19, 163). Moreover, in the same study, the authors described that MCL cells having CCND1 mutations are more resistant towards ibrutinib that the one with no mutation but they did not describe how this mechanism of resistance took place. It is tempting to speculate that the sustained activation of PI3K/AKT or/and NF-κB signaling pathways observed in MCL cells may compensate the inhibition of BTK by ibrutinib. To be more concise the sentence was re-phrased and replaced by : “Moreover, those CCND1 mutations contribute to ibrutinib resistance although this mechanism is still unknown” (page 12, lines 404-405).

Line 369: “The upregulation of MYC and mTORC1 underlies OXPHOS…”

-please re-phrase, so that the reader understands, what is the meaning  of the sentence.

Answer: we agree with the reviewer that this sentence was confusing. We have modified it for the following : “The upregulation of MYC and mTORC1 reprograms the metabolism toward OXPHOS by activating genes involved in glycolysis, glutaminolysis and mitochondrial biogenesis” (page 12, lines 411-413).

Line 378: “Over the past, proteasome inhibition has been demonstrated….”

-should be “Over the past years, proteasome inhibition has been demonstrated…”.

Answer: the correction has been made (page 12, lane 421).

Lines 408-409: MCL-1 interacts with and blocks NOXA, counteracting the ignition of the apoptotic cascade mediated by this BH3-only protein.

-please, re-write. NOXA is a pro-apoptotic sensitizer, not an activator or effector pro-apoptotic protein. By definition, it cannot elicit apoptosis8. Rather, binding of NOXA to MCL1 might trigger apoptosis only in case of displacement of an activator or effector pro-apoptotic player, e.g. BIM, BAX or BAK1 bound to MCL1.

Answer: we acknowledge the reviewer for his/her judicious comment. NOXA is not an apoptosis effector nor an activator. We previously published that in MCL cells, bortezomib induced NOXA accumulation due to proteasome inhibition. NOXA coupled to MCL-1, induced BAK release (Pérez-Galán et al, Blood 2017, 109, 4441). The accumulation of MCL-1 counteracts NOXA-mediated activation of BAK and then the triggering of apoptosis. In turn, blocking NOXA expression as well as inhibiting MCL-1 impacts the apoptotic response. The text has been re-written accordingly and it now appears as follows: “In MCL cells, BTZ leads to the intracellular accumulation of both anti-apoptotic MCL-1 and BH3-only protein, NOXA. By interacting with MCL-1, NOXA allows the release of the pro-apoptotic effector, BAK, leading to mitochondrial depolarization and ignition of the apoptotic cascade” (page 13, lines 450-453).

Lines 423-451: Resistance to lenalidomide.

Lenalidomide mode-of-action in MCL is mediated predominantly through NK Cell-Mediated Cytotoxicity9, 10. The authors should at least comment on that. Mechanisms of resistance to NK-cell-based immunotherapy are nicely reviewed e.g. in the paper of Sordo-Bahamonde et al11.

Answer: we mentioned in the first version of the review the role of immune cells, including NK cells, in the antitumor activity of lenalidomide (ref. 111, Gribben et al, J Clin Oncol 2015, 33, 2803). We agree with the referee that the reference of Hagner et al. should have been included. This error has been corrected by adding this reference (ref 118) in the present revised version. The corresponding sentence has been also modified: “In particular, preclinical studies have shown that lenalidomide enhanced NK cell-mediated cytotoxicity against MCL cells by promoting the formation of lytic immunological synapses and the secretion of granzyme B” (pages 13-14, lines 472-474).

We thank the referee for shedding light on the review by Sordo-Bahamonde et al. Some elements issued from this review have been added in the new version of the manuscript: “Despite these advances, the intrinsic mechanisms of lenalidomide resistance in MCL remain only partially known. Among described mechanisms are the upregulation of MCL-1, the downregulation of BAX and the activation of PI3K/AKT signaling pathway consequently to the interference of the hypoxic TME with NK cell-mediated cytotoxicity [122]. These potential mechanisms are supported by genetic alterations affecting the corresponding genes in MCL [125]” (page 14, lines 483-488).

Line 455: “Deletion of BCL2L11 encoding the pro-apoptotic BIM protein…”.

The data on deletions of BIM in MCL remain controversial. Tagawa et al reported only heterozygous (not homozygous) deletions of BIM in 5 of 27 (18.5%) MCL patients12. Mestre-Escorihuela and colleagues focused mainly on analysis of cell lines and the only information provided about BIM status in primary MCL samples was based on IHC analysis by tissue microarray with loss of BIM protein expression detected in 7 of 22 (33%) patient samples13. Prukova et al analyzed 24 high-risk MIPI patients by aCGH and found only one patient with monoallelic BIM deletion in the context of entire loss of long arms of chromosome 2 within a complex karyotype14. Katz et al focused on genetic proof of concept of biallelic BIM gene deletion during MCL lymphomagenesis, however, did not analyze primary MCL cells15.

Answer: we agree with the reviewer’s comment. The text has been modified to take into account these considerations: “Importantly, homozygous deletion of BIM is mainly observed in MCL cell lines, and the loss of BIM protein found in about one third of MCL patients [126], is unlikely to be explained by the infrequent, heterozygous deletion of the gene reported by Tagawa and coworkers [123], and even not confirmed by others [127]” (page 14, lines 508-512).

Page 591: “vorinostat-BTZ combo has been entered in phase 2 trial”

-please, rephrase.

Answer: the text has been change for the following: …”the combination of vorinostat plus BTZ has been evaluated in a phase 2 trial”… , (page 14, line 659).

Page 593: “Associations with other epigenetic drugs have also been evaluated.”

-please, rephrase… Efficacy of other epigenetic drugs has been evaluated?

Answer: to comply with the second comment of the referee, chapter 4.3 has been substantially modified, and the sentence pointed out in his/her comment now begins by “Beside HDAC and bromodomain inhibition, cladribine, a hypomethylating agent that..” (page 18, line 667).

Round 2

Reviewer 3 Report

The manuscript has been substantially improved.

I do not have any further comments.